# Comparative Phytochemical, Antioxidant, and Hemostatic Studies of Extract and Four Fractions from Paulownia Clone in Vitro 112 Leaves in Human Plasma

**DOI:** 10.3390/molecules25194371

**Published:** 2020-09-23

**Authors:** Weronika Adach, Jerzy Żuchowski, Barbara Moniuszko-Szajwaj, Malgorzata Szumacher-Strabel, Anna Stochmal, Beata Olas, Adam Cieslak

**Affiliations:** 1Department of General Biochemistry, Faculty of Biology and Environmental Protection, University of Łódź, 90-236 Łódź, Poland; weronika.adach@biol.uni.lodz.pl; 2Department of Biochemistry, Institute of Soil Science and Plant Cultivation, State Research Institute, 24-100 Puławy, Poland; jzuchowski@iung.pulawy.pl (J.Ż.); bszajwaj@iung.pulawy.pl (B.M.-S.); asf@iung.pulawy.pl (A.S.); 3Department of Animal Nutrition, Poznań University of Life Sciences, 60-637 Poznań, Poland; malgorzata.szumacher@up.poznan.pl

**Keywords:** plasma, oxidative stress, *Paulownia*, coagulation

## Abstract

**Background:** The Paulownia Clone in Vitro 112, known as oxytree or oxygen tree, is a hybrid clone of the species *Paulownia elongata* and *Paulownia fortunei* (Paulowniaceae). The oxytree is a fast-growing hybrid cultivar that can adapt to wide variations in edaphic and climate conditions. In this work, Paulownia Clone in Vitro 112 leaves were separated into an extract and four fractions (A–D) differing in chemical content in order to investigate their chemical content using LC-MS analysis. The extract and fractions were also evaluated for their anticoagulant and antioxidant properties in a human plasma in vitro. **Results:** The Paulownia leaf extract contained mainly phenolic compounds (e.g., verbascoside), small amounts of iridoids (e.g., aucubin or 7-hydroxytometoside) and triterpenoids (e.g., maslinic acid) were also detected. Our results indicate that the extract and fractions have different effects on oxidative stress in human plasma treated with H_2_O_2_/Fe in vitro, which could be attributed to differences in their chemical content. For example, the extract and all the fractions, at the two highest concentrations of 10 and 50 µg/mL, significantly inhibited the plasma lipid peroxidation induced by H_2_O_2_/Fe. Fractions C and D, at all tested concentrations (1–50 µg/mL) were also found to protect plasma proteins against H_2_O_2_/Fe-induced carbonylation. The positive effects of fraction C and D were dependent on the dose. **Conclusions:** The extract and all four fractions, but particularly fractions C and D, which are rich in phenolic compounds, are novel sources of antioxidants, with an inhibitory effect on oxidative stress in human plasma in vitro. Additionally, the antioxidant potential of fraction D may be associated with triterpenoids.

## 1. Introduction

Paulownia Clone in Vitro 112 (Paulowniaceae), referred to as oxytree or oxygen tree, is a hybrid clone of *Paulownia elongata* and *Paulownia fortunei*. The oxytree is a fast-growing hybrid cultivar that is adaptable to wide variations in edaphic and climatic conditions. The wood of the oxytree is considered hard and can be harvested at least three times throughout the tree′s lifespan. It is not an expansive species, as it does not reproduce generatively [1].

*Paulownia,* and specifically *Paulownia tomentosa* Steud, is widely used in traditional Chinese medicine. It has a broad biological effect, including antioxidant, neuroprotective, anti-inflammatory and antibacterial properties. The bioactive compounds found in *P. tomentosa* include phenylethanoid glycosides, flavonoids, lignans, iridoids, and triterpenoids. The flavonoids protect cells against the toxic effects of glutamate by demonstrating neuroprotective effects. Additionally, they inhibit the activity of acetylcholinesterase and butyrylcholinesterase, which are associated with the alleviation of the symptoms of Alzheimer′s disease. Phenylethanoid glycosides have antioxidant, anti-inflammatory anticancer, hepatoprotective, immunostimulating, and antibacterial. Verabascoside and its several derivatives were isolated from bark and wood of *P. tomentosa* [2,3,4]. *Paulownia* has been shown to contain many bioactive ingredients, e.g., sesamin in the wood, ursolic acid in the leaves and catalpinoside in the bark. Paulownia fruits are rich in, among others, alkaloids and flavonoids with antioxidant properties [3]. The flowers have been found to contain flavonoids such as apigenin, quercetin, apigenin-7-*O*-glucoside, quercetin-3-*O*-glucoside, 3-methoxyluteolin-7-*O*-glucoside, and tricin-7-*O*-glucopyranoside [2]. A comprehensive description of natural products from *P. tomentosa* can be found in a review by Schneiderová and Šmejkal [4].

In this work, Paulownia Clone in Vitro 112 leaves were separated into an extract and four fractions differing in chemical content (A–D). This study aimed to investigate not only the chemical content of the extract and fractions of Paulownia Clone in Vitro 112 leaves, but also their in vitro effects against oxidative stress in human plasma treated with H_2_O_2_/Fe (a donor of hydroxyl radicals). H_2_O_2_ is electrically neutral and readily crosses cell membranes and appears in various parts of cells. In the presence of transition metal ions (mainly Fe^2+^), it undergoes a Fenton reaction, giving a highly reactive hydroxide radical (^•^OH) and a hydroxide anion (OH^−^). ^•^OH reacts with proteins, but mainly initiates the lipid peroxidation process. Hydrogen peroxide is a very good oxidant, it oxidizes the thiol groups of proteins. We used three different parameters of oxidative stress: lipid peroxidation measured by thiobarbituric acid reactive substances (TBARS), thiol group level, and protein carbonylation. Because oxidative stress is correlated with various pathological processes, such as cardiovascular disease, an additional aim of the study was to determine if the extract and fractions could modulate hemostatic parameters of human plasma (including the activated partial thromboplastin time (APTT), the prothrombin time (PT) and the thrombin time (TT)) in vitro. Times of blood clotting were determined coagulometrically.

## 2. Materials and Methods

### 2.1. Chemicals

Dimethylsulfoxide (DMSO), thiobarbituric acid (TBA), and H_2_O_2_ were purchased from Sigma-Aldrich (St. Louis, MO, USA). All other reagents were of analytical grade and were provided by commercial suppliers: POCh (Gliwice, Poland), Chempur (Piekary Slaskie, Poland). ADP and collagen have been purchased from Chrono-Log, Havertown, PA, USA).

#### 2.1.1. Plant Material

Leaves of six-month Paulownia trees were collected from a local plantation at Łęka, Lubelskie Voivodeship, Poland (21°54′ N, 51°27′ E). A voucher specimen (IUNG/PCIV112/2017/1) was deposited at the Department of Biochemistry and Crop Quality, Institute of Soil Science and Plant Cultivation, State Research Institute, Puławy, Poland.

#### 2.1.2. Preparation of the Extract and Fractions

Freshly picked Paulownia leaves were chopped, frozen, and lyophilized (Martin Christ Gamma 2-16 LSC, Germany). Next, the freeze-dried leaves were milled in a laboratory mill (ZM200, Retsch, Haan, Germany) and sieved through a 0.5 mm sieve. The extract of paulownia leaves was obtained through sequential extraction. The powdered plant material was extracted with 5% methanol (*v/v*) using ultrasonic bath, at room temperature for 30 min, and still macerated using a magnetic stirrer for 30 min at room temperature. The content was centrifuged at 4000 g for 10 min. The residue was then re-extracted with 30% methanol (*v/v*) under the same conditions as above, and centrifuged at 4000× *g* for 10 min. Finally, the residue was re-extracted with 70% methanol and centrifuged at 4000× *g* for 10 min. The supernatants were pooled and concentrated under reduced pressure and freeze-dried. The extraction yield was 41.7%.

The crude methanol extract was then purified in a stepwise manner by a range of chromatographic methods. First, the extract was applied to a preconditioned RP-C18 column (80 × 70 mm, 140 µm; Cosmosil C18-PREP; Nacalai Tesque, Kyoto, Japan), and polar constituents were then removed (1% methanol, *v/v*), while active metabolites were eluted with 80% methanol (*v/v*) to give fraction A. The yield of this stage was 44.1%.

Next, fraction A was further separated by flash chromatography on a reversed phase column (140 × 12 mm, 40 μm; Cosmosil C18-PREP; Nacalai Tesque, Kyoto, Japan) connected to a Gilson HPLC apparatus. A linear gradient of aqueous acetonitrile (2–30% *v/v*) containing 0.1% formic acid over 140 min, was used as a mobile phase at a flow rate of 8 mL min^−1^ at ambient temperature. Afterward, fraction A afforded three subfractions B–D. Fraction B constituted 23.4%, fraction C 33.6%, and fraction D 36.3% of fraction A.

#### 2.1.3. LC-MS Analyses

The composition of the extract and fractions was determined by UHPLC-DAD-ESI-MS. Samples were chromatographed using an Acquity UPLC system (Waters, Milford, MA, USA), coupled with an Acquity TQD (Waters) mass detector on an Acquity BEH C18 column (100 × 2.1 mm, 1.7 μm; Waters). The mobile phase was composed of mixtures of solvent A (0.1% formic acid (FA) in Milli-Q water) and solvent B (0.1% FA in acetonitrile). The constituents of the samples were identified on the basis of their MS and UV spectra (identification was supported by spectra and molecular formulas obtained during previously performed LC-HRMS/MS analysis (Q-TOF) of a similar extract from Paulownia Clone in Vitro 112 leaves; data not shown) and literature data. The following elution program was used for semiquantitation of phenolic compounds: 0–1 min: 5% B; 1–14.9 min: 5–40% B (a concave-shaped gradient); 15–17 min: 99% B; 17.10–20 min: 5% B. The column was maintained at 50 °C, the flow rate was 0.500 mL min^−1^, and the injection volume was 2.5 µL. The mass spectrometer was operated in negative and positive ion scanning modes. The following settings were used in negative mode: capillary voltage 2.80 kV, cone voltage 55 V, source temperature 150 °C, desolvation temperature 450 °C, cone gas (N_2_) flow 100 L h^−1^, desolvation gas (N_2_) flow 900 L h^−1^. In positive ion mode, the capillary voltage was 3.10 kV and the cone voltage was 60 V. UV-DAD detection (λ = 330 nm) was used for semiquantitation of phenolic compounds. Levels of individual phenolics were determined using calibration curves of verbascoside (HWI Analytik, Rüelzheim, Germany) and rutin (PhytoLab, Vestenbergsgreuth, Germany). The content of the phenylethanoids, as well as other derivatives of phenolic acids and all minor and unidentified compounds were expressed as equivalent of verbascoside; the content of the major flavonoids was expressed as rutin equivalent.

Iridoids were semiquantified using the following method of elution: 0–2.5 min: 2% B; 2.50–10.0 min: 2–60% B; 10.10–12.10 min: 99% B; 12.20–16 min: 2% B. The column was maintained at 35 °C, the flow rate was 0.400 mL min^−1^, and the injection volume was 2.5 µL. A negative ion SIM method was used: capillary voltage 2.80 kV, cone voltage 30 V, and with the other settings as for phenolics. Two ions were monitored, at *m/z* 407 (FA adduct of catalpol; much higher intensities were observed, as compared to deprotonated ion) and *m/z* 391 (FA adduct of aucubin/7-hydroxytomentoside); dwell times were set automatically. The iridoid content was determined on the basis of a calibration curve of catalpol (Sigma, St. Louis, MO, USA), and expressed as equivalent of catalpol.

The elution program for the semiquantitation of triterpenoids was: 0–0.5 min: 7% B; 0.5–11.90 min: 7–80% B (linear gradient); 12–13 min: 99% B; 13.10–15 min: 7% B. The column was maintained at 50 °C, the flow rate was 0.500 mL min^−1^, and the injection volume was 2.5 µL. A negative ion SIM method was applied: capillary voltage 2.80 kV, cone voltage 80, and other settings as for phenolics. Four ions were monitored (in sequence, within set time ranges), at *m/z* 503, *m/z* 487, *m/z* 471, and *m/z* 455 (chosen on the basis of our preliminary qualitative analyses); the dwell time was 200 ms. The triterpenoid content was determined on the basis of a calibration curve of maslinic acid (Sigma), and expressed as equivalent of maslinic acid.

#### 2.1.4. Preparation of Stock Solutions for Bioassay

Stock solutions of the extract and fractions A–D of Paulownia Clone in Vitro 112 leaves were prepared with 50% (*v/v*) aq. soln. DMSO, a universal solvent for many different plant substances. The final concentration of DMSO in the tested samples was below 0.05% (*v/v*). In addition, as repeatedly confirmed by our earlier studies [5], the addition of a low concentration of DMSO to human plasma has no effect on oxidative stress or coagulation parameters [6].

#### 2.1.5. Isolation of Human Plasma

Human blood and plasma were obtained from non-smoking men and women who were regular donors of a blood bank (RCKiK in Lodz, Poland) and a Medical Center (L. Rydygier Medical Center, in Lodz, Poland). None of the donors had taken any medication, any addictive substances or any antioxidant supplementation. All participants gave informed consent before being inclusion in the study. The study was performed according to the principles given in the Helsinki Declaration. Consent for the study was given by the University of Lodz Bioethical Commission (11/KBBN-UŁ/I/2019).

Plasma was isolated by differential centrifugation as described earlier by Walkowiak et al. [7]. To measure the hemostasis parameters, the plasma was incubated at 37 °C for 30 min with the extract and the four tested fractions (concentration range 1–50 µg/mL). To measure the oxidative stress parameters the plasma was incubated at 37 °C for 30 min with the extract and the four tested fractions (concentration range 1–50 µg/mL) with the addition of 4.7 mM H_2_O_2_/3.8 mM Fe_2_SO_4_/2.5 mM EDTA. The negative control was plasma not treated with H_2_O_2_/Fe, whereas the positive control was plasma treated with H_2_O_2_/Fe.

The protein concentration was calculated according to the procedure devised by Whitaker and Granum [8], on the basis of absorbance measurements at 280 nm (in tested samples).

### 2.2. Parameters of Oxidative Stress

#### 2.2.1. Lipid Peroxidation Measurement

To the test samples after the completed incubation, 500 µL of TCA was added followed by 500 µL of TBA and vortexed for 1 min. Two or three holes were made in eppendorf caps, which were then heated at 100 °C for 10 min. After incubation, samples were cooled for 15 min at 4 °C and centrifuged at 10,000 rpm at 18 °C for 15 min. TBARS (thiobarbituric acid reactive substances) concentration was quantified (at 535 nm wavelength) to assess the degree of oxidative stress. TBARS concentration was calculated using the molar extinction coefficient (ε = 156,000 M^−1^ cm^−1^) as described previously [9,10].

#### 2.2.2. Carbonyl Group Measurement

400 µL 0.9% NaCl was added to 100 µL of the prepared samples. 500 µL of 40% trichloroacetic acid (TCA) was added to the samples and incubated for 5 min to precipitate the protein. After incubation, the samples were centrifuged for 5 min (2500 rpm at 4 °C) and then the supernatant was removed. Next, 750 µL of a 10 mM DNPH solution was added to the pellet, then the samples were shaken for 5 min and incubated at room temperature for 1 h in the dark. After incubation, the samples were placed back on ice and 750 µL of 40% TCA was added to reprecipitate the protein. The samples were centrifuged for 5 min (2500 rpm at 4 °C) and the supernatant removed. 1.5 mL of cooled ethanol-ethyl acetate (1:1) was added to the residue. The samples were centrifuged for 5 min (2500 rpm at 4 °C) and the supernatant removed. This procedure was repeated three times. Then 1 mL of 6 M guanidine hydrochloride in 2 M HCl was added to the pellet and the samples were shaken until the pellet dissolved. The level of carbonyl groups was calculated using the molar extinction coefficient (ε = 22,000 M^−1^ cm^−1^) and was expressed as nmol carbonyl groups/mg of plasma protein [5,10,11].

#### 2.2.3. Thiol Group Measurement

After incubation, the test samples were transferred to a 96-well plate at 20 µL, followed by the addition of 20 µL of SDS and mixed thoroughly. Successively, 160 μL of phosphate buffer (pH 8.0) were added to all samples and mixed thoroughly. The absorbance was measured at a wavelength λ = 412 nm (A_0_) and 16.6 µL DTNB was added. The 96-well plate was incubated for 60 min (temperature 37 °C). After incubation, the absorbance was measured at a wavelength λ = 412 nm (A_1_). The absorbance difference A_1_ − A_0_ was calculated. The level of thiol groups was calculated using the molar extinction coefficient (ε = 13,600 M^−1^ cm^−1^) and was expressed as nmol thiol groups/mg of plasma protein [5,10,12,13].

The level of thiol groups was quantified (at 412 nm wavelength with Ellman′s reagent) to assess the degree of oxidative stress. The level of thiol groups was calculated using the molar extinction coefficient (ε = 13,600 M^−1^ cm^−1^) and was expressed as nmol thiol groups/mg of plasma protein [5,10,12,13].

#### 2.2.4. Parameters of Coagulation

##### The Measurement of Prothrombin Time

The Dia-PT reagent was dissolved in the solvent and incubated at 37 °C for 30 min. 50 μL of plasma was added to the measuring cuvette of the coagulometer and incubated for 2 min on a block heater. Then 100 µL of Dia-PT reagent was added to the measuring cuvette and the coagulometer time measurement was started.

##### The Measurement of Thrombin Time

50 µL of human plasma was added to the measuring cuvette and incubated for 1 min at 37 °C. After incubation, 100 µL of thrombin, dissolved in 0.9% NaCl (final concentration 1 U/mL) was added to the measuring cuvette.

##### The Measurement of APTT

The Dia-PTT reagent was dissolved in 4 mL of 3× distilled water and allowed to dissolve at room temperature for 30 min. In the coagulometer the Dia-CaCl_2_ reagent was heated to 37 °C. 50 μL of plasma and 50 μL of Dia-PTT reagent were added to the measuring cuvette. Plasma and reagent were incubated for 3 min. Then, 50 μL of Dia-CaCl_2_ reagent was added to the measuring cuvette and the time measurement in the coagulometer was started.

Prothrombin time (PT), thrombin time (TT), and activated partial thromboplastin time (APTT) were determined coagulometrically using an Optic Coagulation Analyzer, model K-3002 (Kselmed s.c., Grudziadz, Poland), following the methods described by Malinowska et al. [14].

### 2.3. Data Analysis

Several tests were used to carry out statistical analysis (StatSoft 13.3, TIBCO Software Inc. Palo Alto, CA, USA). To confirm normality, the results obtained were checked using normal probability plots and the equality of variances was verified using the Brown-Forsythe test. Statistically significant differences within and between groups were identified by applying the ANOVA test. This was followed by Duncan′s multiple comparisons test. All values are expressed as mean ± SEM. In order to eliminate uncertain data, the Q-Dixon test was performed. *p*-values below 0.05 were regarded as significant.

## 3. Results

### 3.1. Composition of Paulownia Clone in Vitro 112 Extract and Fractions

Phenolic compounds were the main constituents detected in all the preparations from leaves of Paulownia Clone in Vitro 112 (Table 1). Verbascoside and its derivatives were dominant compounds. The extract and fractions A and B also contained the iridoids, catalpol and aucubin or 7-hydroxytomentoside (Table 2). Small amounts of diverse triterpenoids were additionally found in the extract, fraction A and fraction D. One of them was maslinic acid, identified on the basis of its MS spectrum and retention time, by comparison with the authentic standard (Table 3). Structures of several constituents of the investigated preparations are shown on Figure 1. Chromatograms of the investigated preparations (Appendix A) have been presented in the Appendix A.

### 3.2. Effects on Parameters of Oxidative Stress in Human Plasma

The effects of the extract and the four fractions (concentration range 1–50 µg/mL) on the parameters of oxidative stress in human plasma were studied in an in vitro model. Neither the extract nor any of the fractions altered the level of biomarkers of oxidative stress in plasma not treated with H_2_O_2_/Fe (data not shown). However, exposure of human plasma to the physiological concentrations of H_2_O_2_/Fe induced a significantly increased level of plasma lipid peroxidation, protein carbonylation, and oxidation of protein thiols (Figure 2, Figure 3 and Figure 4). As demonstrated in Figure 1, the extract and all the fractions, at the two highest concentrations of 10 and 50 µg/mL, significantly inhibited the plasma lipid peroxidation induced by H_2_O_2_/Fe (Figure 2). Fractions C and D, at all tested concentrations (1–50 µg/mL) were also found to protect plasma proteins against H_2_O_2_/Fe-induced carbonylation (Figure 3). The positive effects of fraction C and D were dependent on the dose (Figure 2). Moreover, the extract at the two highest concentrations (10 and 50 µg/mL) and all four fractions at all tested concentrations (1, 5, 10, and 50 µg/mL) inhibited the oxidation of plasma protein thiols induced by H_2_O_2_/Fe (Figure 4).

Table 4 compares the effects of the extract and the four fractions at a concentration of 10 µg/mL on the lipid peroxidation and protein carbonylation in plasma treated with H_2_O_2_/Fe. Fractions C and D had stronger anti-oxidant activity than the extract or fractions A and B. For example, the inhibition of protein carbonylation was about 40% for fraction C and about 55% for fraction D (Table 4).

### 3.3. Effects on Hemostatic Parameters of Human Plasma

Analysis of the effect on the coagulation properties of human plasma demonstrated that neither the extract nor any of the fractions in the concentration range 1–50 µg/mL altered PT, the APTT and the TT (Figure 5).

## 4. Discussion

Phenolic compounds were the main constituents detected in all the preparations made from Paulownia Clone in Vitro 112 leaf. Of them, verbascoside (acteoside) was dominant in the extract and fractions A, B, and C, and the other major phenolics were present in much lower amounts (Table 1). Most of the other constituents of these preparations were also phenylethanoid glycosides, as suggested by their MS and UV spectra. Among them, compounds tentatively identified as hydroxyverbascosides and methoxyverbascoside dominated. Compounds of this type (campneoside II, campneoside I, and other), as well as verbascoside, have previously been isolated from the stems and bark of *Paulownia tomentosa* [14], so their presence in Paulownia Clone in Vitro 112 is not surprising. Apart from phenylethanoid glycosides, the extract and fractions A, B, and C also contained significant amounts of apigenin diglucuronide and luteolin diglucuronide. Apigenin-7-*O*-β-d-glucuronopyranosyl (1→2)-*O*-β-d-glucuronopyranoside was found in the bark of *P. tometosa*, so it is probable that the leaves of Paulownia Clone in Vitro 112 contained the same compound. Small amounts of other glycosides of apigenin and luteolin, as well as their aglycones, were also detected. As might be expected, fraction D differed significantly from the remaining preparations. While fractions A, B, and C consisted mainly of verbascoside, hydroxyverbascosides, methoxyverbascoside, and diglucuronides of apigenin and luteolin, fraction D contained a complex mixture of less polar phenolic compounds, poorly separated by the chromatographic method. The UV and MS spectra suggest that these were mainly phenylethanoid glycosides, including acetylverbascoside and dimethylverbsacoside, and possibly other derivatives of hydroxycinamic acids. In addition, fraction D also contained luteolin and apigenin.

The extract and fractions A and B also contained small amounts of two iridoids, namely catalpol and 7-hydroxytomentoside or aucubin. These compounds are also known to be present in the leaves of *P. tomentosa* [14]. 7-hydroxytomentoside/aucubin was the dominant iridoid in all analyzed preparations. The chromatographic peak of the putative 7-hydroxytomentoside/aucubin showed a UV maximum at 249 nm when the samples were analyzed using different UHPLC-MS methods; as 7-hydromentoside, unlike aucubin, has a conjugated system of double bonds in its molecule, it seems to be more probable that the preparations contained mainly 7-hydroxytomentoside. The virtual lack of catalpol in fraction B may be explained by its poor retention on C18 stationary phases: it could be mostly lost during the preparation of the fraction, or alternatively it could decompose.

In addition to phenolics and iridoids, the extract and fractions A and D also contained small amounts of triterpenoids. The extract contained maslinic acid, and different C_30_H_48_O_6,_ C_30_H_48_O_5,_ C_30_H_48_O_4,_ and C_30_H_48_O_3_ compounds, as well as hexosides of C_30_H_48_O_6_. C_30_H_48_O_6_ hexosides and diverse C_30_H_48_O_5_ compounds were major triterpenoids in the extract. Maslinic, colosolic, pomolic (C_30_H_48_O_4_), as well as ursolic, and 3-epiursolic acid (C_30_H_48_O_3_) were earlier found in leaves of *P. tomentosa* [4]. The lack of less polar triterpenoids in fractions A and D was an artifact of the method of preparation of these fractions: 80% methanol was not a sufficiently strong solvent to effectively elute C_30_H_48_O_4_ and C_30_H_48_O_3_ compounds from C18.

The present work is not only the first to characterize the chemical contents of Paulownia Clone in Vitro 112 leaves, but also to describe the effects of the extract and four fractions of these leaves on oxidative stress and hemostasis.

Oxidative stress means an imbalance between oxidants and antioxidants, which leads to redox signaling disorders and damage on the molecular level [15]. Tissue damage results from a series of enzymatic and nonenzymatic reactions that occur in the body. These processes lead to the production of endogenous or exogenous free radicals. Reactive oxygen species (ROS), including hydroxyl radicals (^•^OH), superoxide anions (O_2_^−^) and hydrogen peroxide (H_2_O_2_) are the main causes of oxidative stress. Excess ROS can damage proteins, lipids, DNA, or carbohydrates by altering their structure. They most often form lipid peroxides, reacting with polyunsaturated fatty acids, which lead to cells damage [16,17,18,19].

Low levels of ROS are necessary to control cell growth, cell differentiation, or apoptosis [19]. Small amounts of ROS are produced in aerobic organisms, and are necessary in many biochemical processes present in the body. They are produced in response to external and internal stimuli and are involved in intracellular signal transmission.

Prevention of lipid peroxidation is an essential process in all aerobic organisms, because the products of this process can cause DNA damage. Thiobarbituric acid reactive substances, thiol group oxidation, and carbonyl formation are the most important markers of oxidative stress used at present [20].

Oxidative stress has a significant effect on the process of hemostasis in the human circulatory system and cardiovascular disease (CVD). For example, oxidative stress in the course of chronic diseases, such as diabetes or atherosclerosis, causes excessive activation of plasma thrombin: this action causes increased coagulation, which manifests in laboratory tests by prolonging coagulation times [21,22].

In ideal circumstances, the body deals with mild oxidative stress by regulating the antioxidant defense mechanisms. Antioxidants can delay or prevent oxidative stress by hindering oxidative processes. In chronic diseases, there is an increase in oxidative stress, leading to oxidative damage to biomolecules, including DNA, proteins, and lipids. However, oxidative stress might also be a cause of the primary pathogenic process, not its consequence [23].

Clotting times and markers of oxidative stress are commonly employed to determine the efficacy of novel anti-thrombotic drugs/supplements with antioxidative properties. Blood clotting tests allow to diagnose the causes of blood clots but also to explain the excessive bleeding tendency. In this research, the anti-coagulant effects of the extract and fractions of Paulownia Clone in Vitro 112 leaves were evaluated by measuring APTT, PT and TT in vitro.

PT is the clotting time measurement in the presence of a tissue factor (thromboplastin) as well as calcium ions. This test monitors the efficiency of the extrinsic activity of the plasma coagulation pathway, because the prothrombin time depends on the efficiency of factors V, VII and X of coagulation, as well as prothrombin. In vitro study showed no statistically significant effect of the extract or any fraction (1–50 µg/mL) on prothrombin time, which may mean no effect on prothrombin activity or coagulation factors V, VII and X.

In the case of APTT, the incubation of human plasma with a kaolin suspension activates the coagulation factors XI and XII. The addition of phospholipid (kephalin) and calcium chloride (CaCl_2_) starts the clotting time measurement. The test is performed to analyze the efficiency of the endogenous clotting activation system. The in vitro study showed no statistically significant effect of the extract or any fraction (1–50 µg/mL) on the activated partial thromboplastin time (APTT).

TT is an indicator of the plasma clotting time following the addition of thrombin, an enzyme that converts soluble fibrinogen protein into insoluble fibrin fibers. This time depends mainly on the concentration and properties of fibrinogen, as well as on other factors, including the drugs used. The in vitro study showed no statistically significant effect of the extract or any fraction (1–50 µg/mL) on plasma thrombin time [13].

On the other hand, the extract and all the fractions, particularly fractions C and D, proved to be inhibitors of oxidative stress, as shown by their inhibition of lipid peroxidation, protein carbonylation, and oxidation of thiol groups in human plasma treated with the strong physiological oxidant H_2_O_2_/Fe (a donor of hydroxyl radicals) in an in vitro model. However, the extract and fractions displayed different antioxidant activities in human plasma treated with H_2_O_2_/Fe. For example, at a dose of 50 µg/mL, fractions C and D (which are rich in phenolic compounds, with 741 mg g^−1^ in fraction C and 700 mg g^−1^ in fraction D) exerted stronger inhibitory action on oxidative stress in plasma treated with H_2_O_2_/Fe than did fractions A and B and the extract. For example, the inhibition of lipid peroxidation was about 60% for fraction C. Similar results were observed for other parameters of oxidative stress, including protein carbonylation. Moreover, the antioxidant properties of fraction D may be associated not only with phenolic compounds, but also with triterpenoids.

In summary, we have observed from their inhibitory effect on oxidative stress that the extract and the four fractions (especially the phenolic-rich fractions C and D) are novel sources of antioxidants.

## Figures and Tables

**Figure 1 molecules-25-04371-f001:**
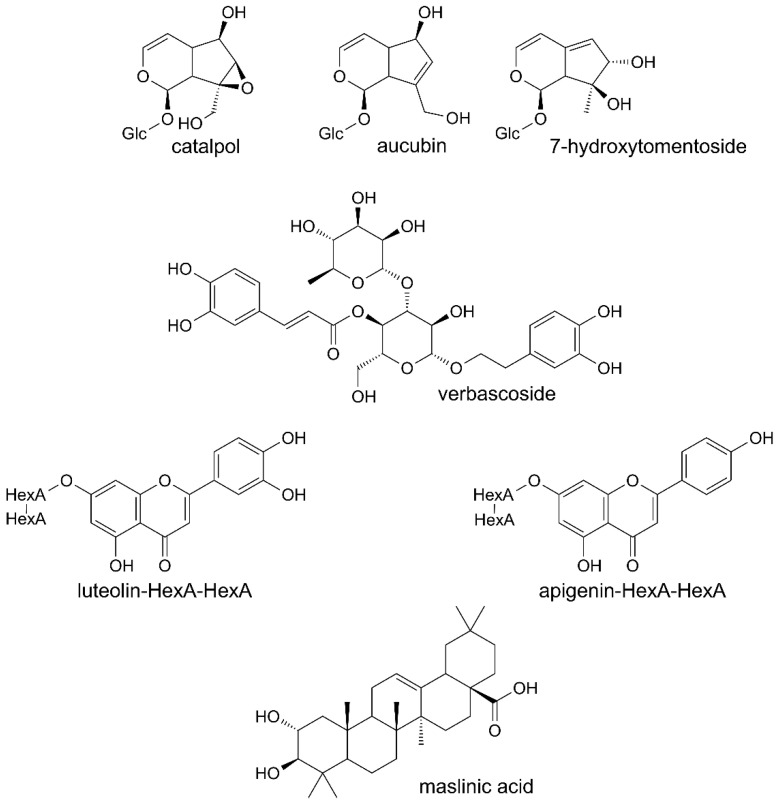
Structures of chosen constituents of the extract and fractions from the leaves of Paulownia Clone in Vitro 112.

**Figure 2 molecules-25-04371-f002:**
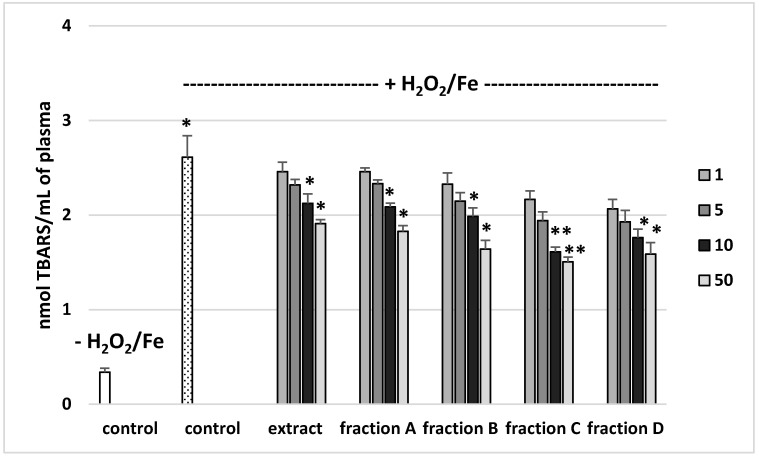
Effects of the extract and the four fractions (A–D, concentration range 1–50 µg/mL, pre-incubation time—5 min) from Paulownia Clone in Vitro 112 leaves on lipid peroxidation in plasma treated with H_2_O_2_/Fe (incubation time—25 min). Results are given as mean ± SEM (*n* = 6). Control negative (white bar) refers to plasma not treated with H_2_O_2_/Fe, whereas control positive (dot bar) to plasma treated with H_2_O_2_/Fe. One-way ANOVA followed by a multicomparison Duncan′s test: * *p* ≤ 0.05, ** *p* ≤ 0.02, compared with positive control (treated with H_2_O_2_/Fe); * *p* ≤ 0.05, compared between negative control and positive control.

**Figure 3 molecules-25-04371-f003:**
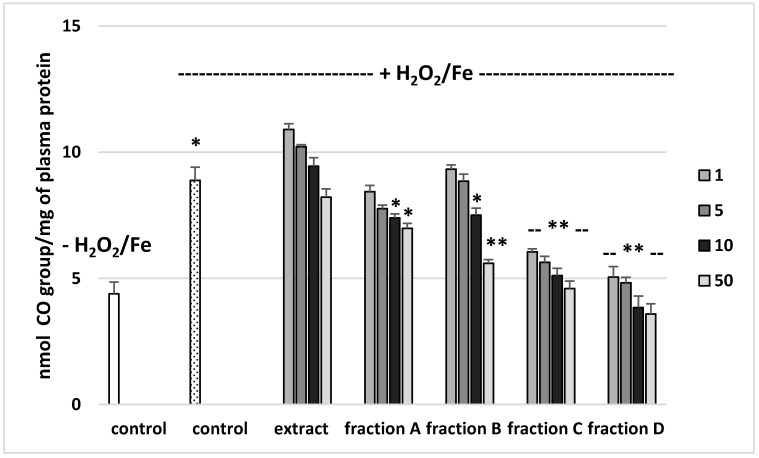
Effects of the extract and the four fractions (A–D, concentration range 1–50 µg/mL, pre-incubation time—5 min) from Paulownia Clone in Vitro 112 leaves on the oxidative damages of plasma protein treated with H_2_O_2_/Fe (incubation time—25 min)-protein carbonylation. Results are given as mean ± SEM (*n* = 6). Control negative (white bar) refers to plasma not treated with H_2_O_2_/Fe, whereas control positive (dot bar) to plasma treated with H_2_O_2_/Fe. One-way ANOVA followed by a multicomparison Duncan’s test: -- *p* > 0.05, * *p* ≤ 0.05, ** *p* ≤ 0.02, compared with positive control (treated with H_2_O_2_/Fe); * *p* ≤ 0.05, compared between negative control and positive control.

**Figure 4 molecules-25-04371-f004:**
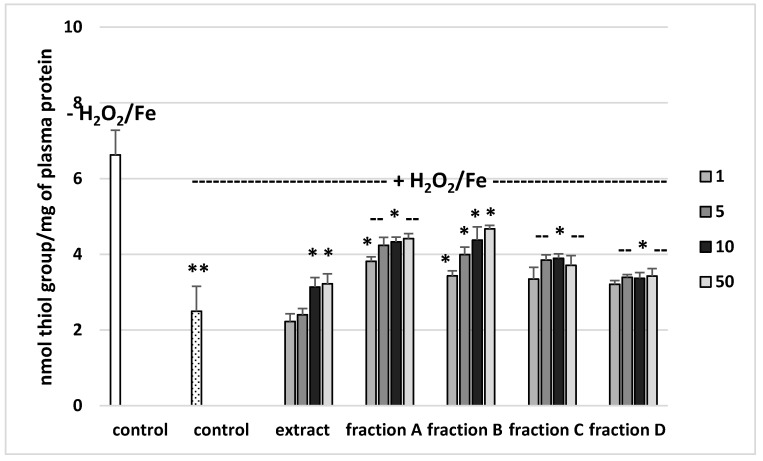
Effects of the extract and the four fractions (A–D, concentration range 1–50 µg/mL, pre-incubation time—5 min) from Paulownia Clone in Vitro 112 leaves on the oxidative damages of plasma protein treated with H_2_O_2_/Fe (incubation time—25 min)-the level of thiol groups. Results are given as mean ± SEM (*n* = 6). Control negative (white bar) refers to plasma not treated with H_2_O_2_/Fe, whereas control positive (black bar) to plasma treated with H_2_O_2_/Fe. One-way ANOVA followed by a multicomparison Duncan′s test: -- *p* > 0.05, * *p* ≤ 0.05, ** *p* ≤ 0.02, compared with positive control (treated with H_2_O_2_/Fe).

**Figure 5 molecules-25-04371-f005:**
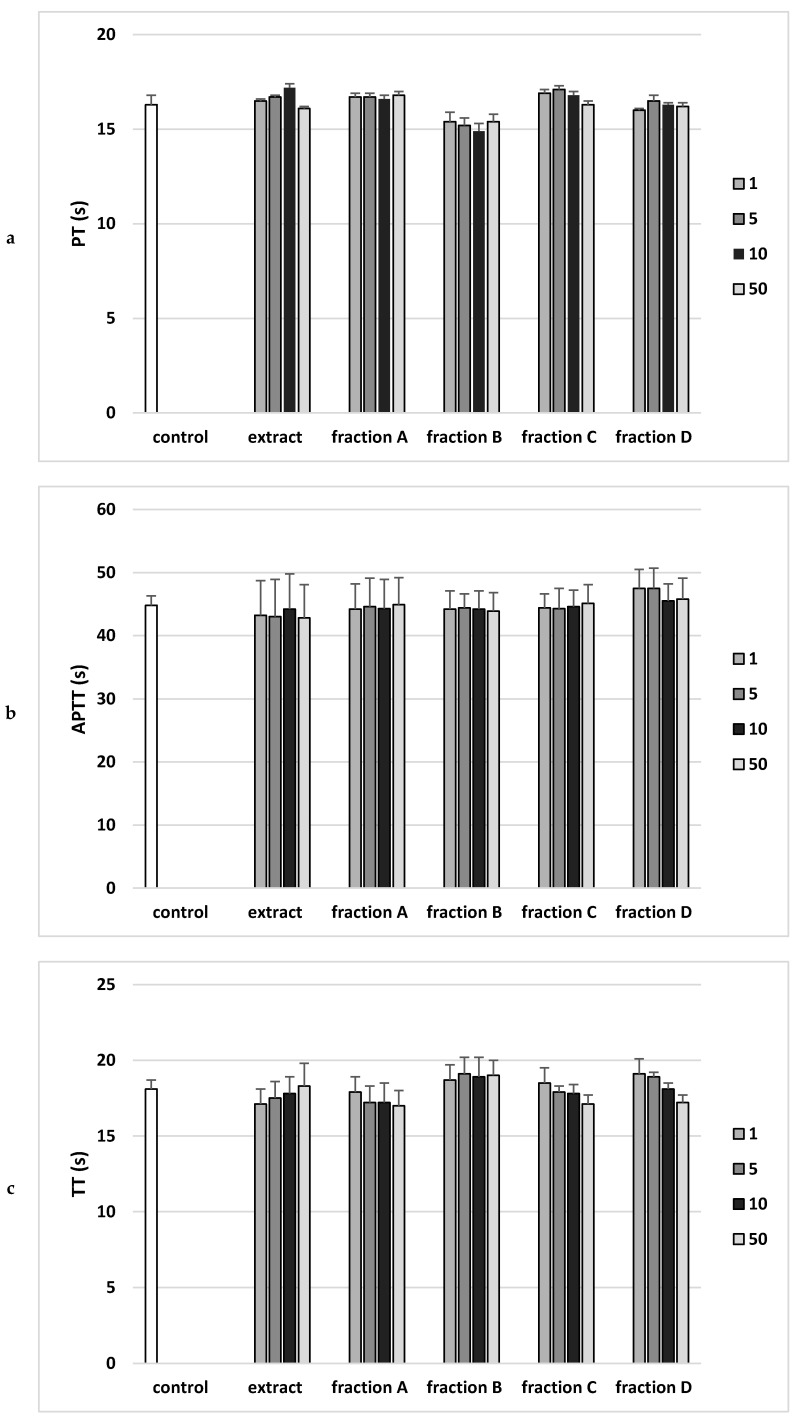
Effects of the extract and the four fractions (A–D, concentration range 1–50 µg/mL, incubation time—30 min) from Paulownia Clone in Vitro 112 leaves on the hemostatic parameters: (**a**) prothrombin time (PT), (**b**) activated partial thromboplastin time (APTT), and (**c**) thrombin time (TT) of human plasma. Data are expressed as mean ± SEM (*n* = 6). One-way ANOVA followed by a multicomparison Duncan’s test: *p* > 0.05, compared with control.

**Table 1 molecules-25-04371-t001:** Content of phenolic compounds in the extract from Paulownia Clone in Vitro 112, and its fractions.

Compound	Phenolic Compounds (mg·g^−1^ ± SD)				
	Extract	Fraction A	Fraction B	Fraction C	Fraction D
caffeic acid-Hex-dHex	2.7 ± 0.10 *	6.9 ± 0.08 *	26.8 ± 1.86 *		
luteolin-HexA-HexA	6.9 ± 0.47 ^$^	17.0 ± 0.75 ^$^	28.4 ± 1.68 ^$^	38.1 ± 1.44 ^$^	Traces
hydroxyverbascoside	7.2 ± 0.23 *	16.2 ± 0.22 *	31.8 ± 2.04 *	28.0 ± 1.01 *	1.1 ± 0.01 *
hydroxyverbascoside	7.9 ± 0.27 *	17.9 ± 0.34 *	34.3 ± 2.11 *	31.3 ± 1.17 *	1.3 ± 0.03 *
apigenin-HexA-HexA	14.2 ± 0.45 ^$^	30.7 ± 0.27 ^$^	31.0 ± 1.88 ^$^	80.5 ± 3.20 ^$^	Traces
methoxyverbascoside	17.4 ± 0.65 *	36.4 ± 0.41 *	45.6 ± 2.83 *	84.3 ± 2.91 *	1.5 ± 0.10 *
verbascoside	74.6 ± 4.88	164.9 ± 1.81	154.1 ± 9.94	424.7 ± 14.23	4.5 ± 0.10
other phenolics	74.5 ± 5.90 *	163.7 ± 1.73 *	26.9 ± 2.03 *	56.2 ± 4.21 *	691.7 ± 5.73 *
total phenolics	205.5 ± 6.41	453.7 ± 5.22	379.0 ± 24.25	743.1 ± 25.96	700.0 ± 5.76

Hex—hexose; dHex—deoxyhexose; HexA—hexuronic acid; * equivalent of verbascoside; ^$^ equivalent of rutin.

**Table 2 molecules-25-04371-t002:** Content of iridoids in the extract from Paulownia Clone in Vitro 112, and its fractions.

Compound	Iridoids (mg·g^−1^ ± SD)		
	Extract	Fraction A	Fraction B
Catalpol	0.66 ± 0.007	1.52 ± 0.046	Traces
7-hydroxytomentoside/aucubin	14.50 ± 0.271 *	25.41 ± 0.440 *	19.77 ± 0.415 *
total iridoids	15.16 ± 0.274 *	26.93 ± 0.484 *	19.77 ± 0.415 *

* equivalent of catalpol.

**Table 3 molecules-25-04371-t003:** Content of triterpenoids in the extract from paulownia Clone in Vitro 112, and its fractions.

Compounds	Triterpenoids (mg·g^−1^ ± SD)		
	Extract	Fraction A	Fraction D
total C_30_H_48_O_6_-Hex	1.36 ± 0.015 *^,#^	2.82 ± 0.015 *^,#^	7.19 ± 0.223 *^,#^
total C_30_H_48_O_6_	0.19 ± 0.033 *	0.41 ± 0.031 *	1.65 ± 0.023 *
total C_30_H_48_O_5_	1.15 ± 0.130 *	0.99 ± 0.076 *	0.53 ± 0.010 *
maslinic acid (C_30_H_48_O_4_)	0.33 ± 0.044	traces	Traces
other C_30_H_48_O_4_	0.54 ± 0.060 *	traces	Traces
total C_30_H_48_O_3_	0.08 ± 0.007 *		
total triterpenoids	3.65 ± 0.278	4.23 ± 0.121	9.36 ± 0.010

Hex—hexose; * equivalent of maslinic acid; ^#^ understated value.

**Table 4 molecules-25-04371-t004:** Comparative effects of Paulownia Clone in Vitro 112 extract and the four fractions (A–D) at the tested dose (10 µg/mL) in selected biomarkers of oxidative stress in plasma treated with H_2_O_2_/Fe.

Type of Extract and Fraction from Paulownia Clon In Vitro 112 Leaves	Inhibition of Lipid Peroxidation (%)	Inhibition of Protein Carbonylation (%)
Extract (a)	18.7 ± 6.7	-
Fraction A (b)	20.1 ± 5.5 (*p* > 0.05 b vs. a)	16.7 ± 3.4 (*p* < 0.05 b vs. a)
Fraction B (c)	23.9 ± 4.9 (*p* > 0.05 c vs. a, b)	15.4 ± 7.2 (*p* < 0.05 c vs. a; *p* > 0.05 c vs. b)
Fraction C (d)	38.2 ± 4.4 (*p* < 0.05 d vs. a, b, c)	42.4 ± 8.9 (*p* < 0.05 d vs. a, b, c)
Fraction D (e)	32.2 ± 5.2 (*p* < 0.05 e vs. a, b, c; *p* > 0.05 e vs. d)	56.7 ± 10.4 (*p* < 0.05 e vs. a, b, c; *p* > 0.05 e vs. d)

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
