# Peer review of "Comparative Phytochemical, Antioxidant, and Hemostatic Studies of Extract and Four Fractions from Paulownia Clone in Vitro 112 Leaves in Human Plasma"

_molecules, 2020, doi:10.3390/molecules25194371_

Round 1

Reviewer 1 Report

The article is interesting, especially since there are not many studies that refer to the biological importance of products obtained from this species.

However, the authors need to make several additions for a scientific growth of the article.

The Introduction should be supplemented with other studies that have been done to establish  the phytochemical profile of Paulownia Clone in vitro 112 and the importance of these compounds for medicine. This in the context in which the authors state in lines 275-276 that “The present work is not only the first to characterize the chemical contents of Paulownia Clone in Vitro 112 leaves”.

In Results, the tables must be redone, because they are very difficult to follow as the authors did, the numbering of the rows mixing with the data presented in the tables.

Returning to the authors' statement in lines 275-276, please compare the results with the other studies and highlight the amount of scientific information resulting from this study.

It is very important to present the chromatograms to the Results, at least as Supplementary materials.

In Discussions line 264 what is the chemical structure of 7-hydromentoside?

It would be very useful if the chemical structures of the important compounds highlighted in each fraction were shown in a figure.

The citation of the references must be redone in accordance with the MDPI rules, using square brackets. The list of references also needs to be revised, a series of meaningless additional numberings appear in the text.

A number of other small corrections have been reported in the text of the attached article.

Author Response

importance of products obtained from this species.

However, the authors need to make several additions for a scientific growth of the article.

The Introduction should be supplemented with other studies that have been done to establish  the phytochemical profile of Paulownia Clone in vitro 112 and the importance of these compounds for medicine. This in the context in which the authors state in lines 275-276 that “The present work is not only the first to characterize the chemical contents of Paulownia Clone in Vitro 112 leaves”.

Response: Unfortunately we have not been able to find any publication concerning phytochemical composition of Paulownia clone in Vitro 112 during our literature search. However, P. tomentosa was broadly investigated in respect of its composition and biological activity, and some information on this subject were included in the Introduction

In Results, the tables must be redone, because they are very difficult to follow as the authors did, the numbering of the rows mixing with the data presented in the tables.

Response: This most unfortunate defect has been removed in the revised manuscript.

Returning to the authors' statement in lines 275-276, please compare the results with the other studies and highlight the amount of scientific information resulting from this study.

Response: We have added more information about it in the chapter of Introduction: “In this work, Paulownia Clone in Vitro 112 leaves were separated into an extract and four fractions differing in chemical content (A–D). This study aimed to investigate not only the chemical content of the extract and fractions of Paulownia Clone in Vitro 112 leaves, but also their in vitro effects against oxidative stress in human plasma treated with H2O2/Fe (a donor of hydroxyl radicals). We used three different parameters of oxidative stress: lipid peroxidation measured by thiobarbituric acid reactive substances (TBARS), thiol group level, and protein carbonylation. Because oxidative stress is correlated with various pathological processes, such as cardiovascular disease, an additional aim of the study was to determine if the extract and fractions could modulate hemostatic parameters of human plasma (including the activated partial thromboplastin time (APTT), the prothrombin time (PT) and the thrombin time (TT)) in vitro. Times of blood clotting were determined coagulometrically.” However, we have added other information, because there is no date about it at other papers. Our manuscript is first, which describes biological activity.

It is very important to present the chromatograms to the Results, at least as Supplementary materials.

Response: A Supplementary materials file has been prepared, containing UV chromatograms of all investigated preparations, as well as a base peak MS chromatogram of the crude extract.

In Discussions line 264 what is the chemical structure of 7-hydromentoside?

Response: The mentioned fragment has been corrected.

It would be very useful if the chemical structures of the important compounds highlighted in each fraction were shown in a figure.

Response: A figure showing structures of several detected constituents of the investigated preparations has been added to the manuscript. We know too little about structures of the tentatively identified derivatives of verbascoside, so they were not be presented in the figure.

The citation of the references must be redone in accordance with the MDPI rules, using square brackets. The list of references also needs to be revised, a series of meaningless additional numberings appear in the text.

Response: We have corrected.

A number of other small corrections have been reported in the text of the attached article.

Reviewer 2 Report

This paper describes components of methanol extract of  Paulownia Clone in Vitro 112, known as oxytree or oxygen tree. And anti-oxidant activity. The extract and fractions which rich in phenolic compounds showed antioxidant effect.

 This tree is not familiar and not so many papers.   

Q1. Why did not extract with ethyl acetate and hexan after methanol extract?

Q2. Why dod not show the chromatographic chart of LC-MS. How many peaks were found? Are there any other compound other than phenolic compounds ?

Q3. Why fraction A was further fractionated to fraction B,C and D. What is the aim? Why removed polar constituents (L.81)

Q4. Why did you chose m/z of 407, 391, 503, 487, 471 and 455? (L.114, L.121)

Q5. Crude extract and fraction A,B,C and D may have color. Does these color interfere the TBARS and other measurements, since TBARS uses absorbance at 535nm.

Minor points.

Q6. In abstract, add one or two components found. 

Q7. In fig 2, (L.205) 1 micro g of extract value is larger than control. Why? Also in fig 3, 1 micro g of fraction A larger than control. Why ?  If this depend on sample, 5, 10 and 50 increased more, but not so. It is strange.

Author Response

Rev. 2

his paper describes components of methanol extract of  Paulownia Clone in Vitro 112, known as oxytree or oxygen tree. And anti-oxidant activity. The extract and fractions which rich in phenolic compounds showed antioxidant effect.

This tree is not familiar and not so many papers.   

Q1. Why did not extract with ethyl acetate and hexan after methanol extract?

Response: Of course, it could be done and it would result in some kind of fractionation of the extract.  However, it is not obvious that it would give a better result. But we decided for C18 SPE. In this way we got rid sugars, organic acids and other highly polar compounds (Fr. A).  Further fractionation turned out not to be very efficient. However, we still obtained a fraction containing iridoids and major phenolics (Fr. B), a fraction of major phenolics (Fr. C) and a fraction of less polar phenolics and triterpenoids.

Q2. Why dod not show the chromatographic chart of LC-MS. How many peaks were found? Are there any other compound other than phenolic compounds ?

Response: Information about some non-phenolic constituents, i.e. iridoids and triterpenoids, can be found in the initial version of the manuscript, actually. Some other compounds were also found, both more and less polar, but they were not identified; such compounds seldom gave any greater MS peaks in LC-MS analyses performed for this paper. Regardless of these observations, phenolic compounds seemed to constitute a big majority of all secondary metabolites present in the extract. A Supplementary material file has been prepared, showing a negative ion base peak chromatogram of the extract, as well as UV chromatograms of all investigated preparations.

Q3. Why fraction A was further fractionated to fraction B,C and D. What is the aim? Why removed polar constituents (L.81)

Response: The fraction A was further fractionated to obtain preparations with different composition of secondary metabolites. Our fractionation turned out not to be very effective, I am afraid. Nevertheless, we obtained a fraction containing iridoids and major phenolics (Fr. B), a fraction of major phenolics (Fr. C), and a fraction of less polar phenolics and more polar triterpenoids. As regards the removal of polar constituents – our study was focused on secondary metabolites, so we wanted to get rid of sugars and organic acids.

Q4. Why did you chose m/z of 407, 391, 503, 487, 471 and 455? (L.114, L.121)

Response: We knew from our preliminary LC-MS (both low resolution and high resolution) analyses, supported by literature about Paulownia tomentosa, that the extract contained probably catalpol, aucubin or 7-hydroxytomentoside, and tritepenoids giving deprotonated ions at m/z 455, 471, 487, and 503. Intensities of formic acid adduct ions of catalpol and aucubin/7-hydroxytmomentoside (m/z 407 and m/z 391, respectively) were much higher than those of deprotonated ions, so we decided to detect the adduct ions to obtain a better sensitivity.

Q5. Crude extract and fraction A,B,C and D may have color. Does these color interfere the TBARS and other measurements, since TBARS uses absorbance at 535nm.

Response: We have done different control and we have not observed interfere……

Minor points.

Q6. In abstract, add one or two components found. 

Response: They have been added to the revised abstract

Q7. In fig 2, (L.205) 1 micro g of extract value is larger than control. Why? Also in fig 3, 1 micro g of fraction A larger than control. Why ?  If this depend on sample, 5, 10 and 50 increased more, but not so. It is strange.

Response: We have observed that 1 micro g of extract value is larger than control, however this increase was not statistically significant. The same process was for 1 micro g of fraction A larger than control (Fig. 3).

Reviewer 3 Report

These are my comments on the referenced manuscript.

The abstract needs to be re-organized. The Results section does not show actual results. A lot of space is used for Background and Methods.
Please check the tables. Those have numbers all over that need to be removed.
The data in Table 1 is from biological or technical replicates?
Table 2. 65. Please add the word "Mean" after the units.
In the Methods section, the authors mentioned that a p-value lower than 0.05 was significant. Therefore, in Figure Legends omit the use of other p-values. That is useless.
In figures 1, 2, and 3, please label X-axis with "Treatment"

Author Response

Response: We have corrected the abstract, the results and other.

Reviewer 4 Report

The study is of interest, however some issues should be better addressed.

Specific comments: 

1) All the procedures including sample preparation and treatments in the subsection 2.2.4 should be specified

2) Results from the effect of the extract and fractions on clotting times should be discussed in details.

Author Response

Specific comments: 

1) All the procedures including sample preparation and treatments in the subsection 2.2.4 should be specified

Response: We have added more information about sample preparation and treatments in the subsection 2.2.4.

2) Results from the effect of the extract and fractions on clotting times should be discussed in details.

Response: We have added more information about the effect of the extract and fractions on clotting times in the Discussion section.

Reviewer 5 Report

In this manuscript, by Adach et al, is described the preparation of an extract and 4 fractions from Paulownia Clone in Vitro 112 leaves, and their partial phytochemical characterization by LC-MS. In addition, the antioxidant and hemostatic effects of these 5 different products were also partially studied. Interesting results were observed concerning the antioxidant properties which can be related with the different composition of these 5 different products.

This is an interesting study with potential future interest, which can allow a valorization of this fast-growing hybrid cultivar. I consider that this study is relevant to publish, however, there are some improvements that must be performed:

-other antioxidant assays should be included in the study (e.g. DPPH and others), because only one type of antioxidant assay in this context generally is little

-it would also be very interesting to include cytotoxicity studies concerning the effects of the 5 products in human cell lines

Over all text (including figures and tables):

-“Paulownia Clone in Vitro 112” designation must be standardized (for example in legends we can find “paulownia CLON IN VITRO 112”; the same for Paulownia (sometimes is in italics, sometimes appears starting with a capital letter…)

Abstract:

-line 23 “in a human plasma in in vitro model”?

-the Results subsection must be improved, namely by including more specific results (some of these are in the Conclusions subsection)

Introduction:

-sentence in lines 37/38 must be improved

-the reference in line 42 is not adequate for the corresponding text

-the second paragraph, concerning the bioactive compounds, is confuse and therefore must be improved. In addition, refer that “The bioactive compounds in Paulownia are flavonoids, lignans, and iridoids” is not completely correct

-line 51: “…treated with H2O2/Fe (a donor of hydroxyl radicals)…” – this oxidative system not only produces radical hydroxyl and therefore, a deeper explanation on the reactions involved and compounds originated should be included. In addition, in line 187, the authors referred that this oxidant system is physiological – this also must be clarified in the text

Materials and methods:

-lines 94/95 – FA?

-sentence in lines 95-97 – the given information must be completed and clarified

-line 128 – “our earlier studies” – please include the adequate references

-an isolated compound with recognized antioxidant properties and anticoagulant effects should be included in the studies as a clear positive control

-the assays in section 2.2 must be adequately described

-section 2.2: I could not find references Bartosz, 2008 and Olas et al. 2018 in the references list

-sentence in line 162?

Discussion:

-line 274 – weak solvent?

-line 282 – please standardize the symbol of radical

-how to explain the lack of effect on coagulation by these 5 products? Is this positive or negative?

References:

-must be numbered in order of appearance in the text

-references list is confuse – generally the numbers do not correspond to a given document

-the reference “Bradford, M.M., 1976” is in the list but it was not present in the text…

Author Response

This is an interesting study with potential future interest, which can allow a valorization of this fast-growing hybrid cultivar. I consider that this study is relevant to publish, however, there are some improvements that must be performed:

-other antioxidant assays should be included in the study (e.g. DPPH and others), because only one type of antioxidant assay in this context generally is little

Response: We have plan to study the antioxidant potential of tested extracts using other tests, including DPPH.

-it would also be very interesting to include cytotoxicity studies concerning the effects of the 5 products in human cell lines

Response: Our preliminary results have demonstrated that neither the extract nor any of the fractions caused lysis of platelets at any dose (1–50 µg/mL). We want to publish these results in other manuscript – tested plant preparations and blood platelets.

Over all text (including figures and tables):

-“Paulownia Clone in Vitro 112” designation must be standardized (for example in legends we can find “paulownia CLON IN VITRO 112”; the same for Paulownia (sometimes is in italics, sometimes appears starting with a capital letter…)

Response: It has been unified in the whole text.

Abstract:

-line 23 “in a human plasma in in vitro model”?

Response: We have corrected. Now, it is only “in vitro”

-the Results subsection must be improved, namely by including more specific results (some of these are in the Conclusions subsection)

Response: We have added more information in the results and the conclusions.

Introduction:

-sentence in lines 37/38 must be improved

Response: We have corrected.

-the reference in line 42 is not adequate for the corresponding text

Response: We have corrected.

-the second paragraph, concerning the bioactive compounds, is confuse and therefore must be improved. In addition, refer that “The bioactive compounds in Paulownia are flavonoids, lignans, and iridoids” is not completely correct

Response: The introduction has been modified, the current version should be better.

-line 51: “…treated with H2O2/Fe (a donor of hydroxyl radicals)…” – this oxidative system not only produces radical hydroxyl and therefore, a deeper explanation on the reactions involved and compounds originated should be included. In addition, in line 187, the authors referred that this oxidant system is physiological – this also must be clarified in the text

Response: We have added more information about the reactions involved and compounds originated.

Materials and methods:

-lines 94/95 – FA?

Response: Formic acid – the abbreviation has been explained in the revised manuscript

-sentence in lines 95-97 – the given information must be completed and clarified

Response: The description has been modified. I hope that the revised version is sufficiently clear

-line 128 – “our earlier studies” – please include the adequate references

Response: We have added reference – [5].

-an isolated compound with recognized antioxidant properties and anticoagulant effects should be included in the studies as a clear positive control

Response: We always prepare the positive control f.e. quercetin, kaempferol or resveratrol which have antioxidant properties. Moreover, these compounds modulate parameters of hemostasis (Rolnik et al., ICP2020). In our present studies, we have alo done experiemnts with these controls. However, we have not added these results.

-the assays in section 2.2 must be adequately described

Response: We have corrected.

-section 2.2: I could not find references Bartosz, 2008 and Olas et al. 2018 in the references list

Response: We have corrected.

-sentence in line 162?

Response: We have corrected.

Discussion:

-line 274 – weak solvent?

Response: this phrase has been replaced by a better one.

-line 282 – please standardize the symbol of radical

Response: We have corrected.

-how to explain the lack of effect on coagulation by these 5 products? Is this positive or negative?

Response: The tested extract and fractions from Paulownia CLON IN VITRO 112 did not show any effect on the coagulation system, therefore their activity can be considered neutral. The detailed effect of the extract and the fraction on the clotting times is presented in the discussion section, paragraph 10. Moreover, tested plant preparations may not be a good source of active substances – procoagulants and anticoagulants for pharmacological applications. However, more extensive studies are needed to determine.

References:

-must be numbered in order of appearance in the text

Response: We have corrected.

-references list is confuse – generally the numbers do not correspond to a given document

Response: We have corrected.

-the reference “Bradford, M.M., 1976” is in the list but it was not present in the text…

Response: We have corrected.

Round 2

Reviewer 1 Report

I consider that the authors took into account the observations and the scientific quality of the article increased.
Small corrections still need to be made, in the sense that the authors must use uniformly the expression of the units of measurement.

instead of µl  use µL

instead of ml use  mL

Reviewer 2 Report

It is interesting about Oxytree components.

Reviewer 5 Report

The authors answered the majority of the previously raised points/questions and therefore the text was improved and now is more acceptable for publication. However, several mistakes still can be detected and it is suggested a new and detailed analysis of the text to improve it - examples of decected mistakes:

-the English should be improved (ex: sentence in lines 18-20)

-"CLON IN VITRO 112" in Table 4 - again, please uniformize in all text

-"in vitro" in italics (line 337) - please uniformize in all text